# A Sensitive and Flexible Capacitive Pressure Sensor Based on a Porous Hollow Hemisphere Dielectric Layer

**DOI:** 10.3390/mi14030662

**Published:** 2023-03-16

**Authors:** Haoao Cui, Yijian Liu, Ruili Tang, Jie Ren, Liang Yao, Yuhao Cai, Da Chen

**Affiliations:** 1Laboratory for Intelligent Flexible Electronics, College of Electronic and Information Engineering, Shandong University of Science and Technology, Qingdao 266590, China; 2Beijing Smart-Chip Microelectronics Technology Co., Ltd., Beijing 100192, China

**Keywords:** capacitive pressure sensors, salt template method, porous structure, groove, high sensitivity

## Abstract

Capacitive pressure sensors based on porous structures have been widely researched and applied to a variety of practical applications. To date, it remains a big challenge to develop a capacitive pressure sensor with a high sensitivity and good linearity over a wide pressure range. In this paper, a sensitive, flexible, porous capacitive pressure sensor was designed and manufactured by means of the “salt template method” and man-made grooves. To this aim, the size of the salt particles used for forming pores/air voids, time taken for thorough dissolution of salt particles, and the depth of the man-made groove by a pin were taken into consideration to achieve a better effect. With pores and the groove, the sensor is more liable be compressed, which will result in a dramatic decrease in distance between the two electrodes and a conspicuous increase of the effective dielectric constant. The optimize-designed sensor represents a sensitivity 6–8 times more than the sensor without the groove in the pressure range of 0–10 kPa, not to mention the sensor without pores or the groove, and it can keep good linearity within the measurement range (0–50 kPa). Besides, the sensor shows a low detection limit of 3.5 Pa and a fast response speed (≈50 ms), which makes it possible to detect a tiny applied pressure immediately. The fabricated sensor can be applied to wearable devices to monitor finger and wrist bending, and it can be used in the object identification of mechanical claws and object cutting of mechanical arms, and so on.

## 1. Introduction

Flexible pressure sensors based on polymers have been widely applied in human health monitoring [1], motion detection [2], and wearable devices [3]. Among various types of pressure sensors, such as capacitive [4], piezoresistive [5], piezoelectric [6], and triboelectric sensors [7], capacitive pressure sensors have been extensively studied and employed in terms of their simple design, excellent stability, and fast dynamic response [8]. Capacitive pressure sensors are composed of a deformable dielectric material sandwiched between two flexible conductive electrodes [9]. Polydimethylsiloxane (PDMS) [10,11], thermoplastic polyurethane (PU) [11,12], and Ecoflex [12] are ideal and conventional elastomers to prepare the dielectric layer because of their low elasticity modulus [13]. Besides, many conductive materials, such as two-dimensional materials [14,15], nanowires [16], metallic materials [17], and polymer gels [18], are involved in the fabrication of sensor electrodes [19]. The development and utilization of new materials brings new opportunities for flexible pressure sensors. The working principle of capacitive pressure sensors can be briefly described as follows: when an external pressure is applied to the flexible pressure sensor, the thickness and relative permittivity of the dielectric layer varies, which can result in a capacitance change of the sensor.

Till now, dielectric layer has been microstructured into different shapes, [20] such as cones [21], pyramids [22], pillars [23], porous structures, and rough surfaces [24], so as to improve the sensitivity of the flexible capacitive sensor. Many such microstructures are prepared by using traditional three-dimensional (3D) printing [21], e-beam evaporation [25], and lithographic methods [26,27]. Although the fabricated microstructures are accurate and controllable, [28] the manufacturing process of such technologies are relatively complex and high-cost, which severely interferes with its widespread application. Exploiting natural existing biomaterials such as lotus leaves [3] and mimosa leaves as templates to directly manufacture the microstructures can be a simple and cost-effective substitute approach. Unfortunately, the as-fabricated patterns are prone to be bound by the innate morphology of the natural biomaterials, leading to uncontrollable shape, dimension, and spacing. Flexible pressure sensors based on microstructured dielectric layers possess a low elasticity modulus and viscoelasticity, so their sensing performance is superior to flexible pressure sensors based on solid dielectric layers [11]. However, it should be noted that the capacitive sensors based on the two abovementioned fabrication methods have a low-pressure range since patterns collapse quickly even under low pressure [29,30]. In this context, the particle template method [31] is a commonly used strategy to prepare a proteiform elastomer simply and cheaply with favorable reproducibility. Solid particles which will be dissolved by the proper solvent to form pores in the elastomer, such as polystyrene (PS) beads [22], salt [31], and sugar [4], are widely used as sacrifice materials. In view of the processing cost and environmental sustainability, the salt/sugar dissolution method [32] is often the most popular choice. For example, Young Junget et al. [33] prepared a highly sensitive and flexible capacitive pressure sensor based on a porous three-dimensional PDMS composited by cubed sugar. The porous structured pressure sensor featured a high sensitivity of 0.124 kPa^−1^ in a range of 0–15 kPa, with a fast rise time (~167 ms) and fall time (~117 ms), as well as excellent reproducibility. As for the drawbacks of the “salt template method”, the discrete particles are more liable to be completely wrapped by the PDMS, leading to a slow dissolution speed, large particle residue, and low pore interconnectivity. Besides, higher sensitivity is needed to be explored further.

Here, we report a flexible capacitive pressure sensor based on a dielectric PDMS elastomer with pores and a man-made groove, which achieves a relatively high sensitivity and favorable linearity in a wide pressure range. According to our experimental experience, we found a more appropriate mass ratio (PDMS:Salts = 2:9) to effectively reduce salt residues and achieve high interconnectivity inside the elastomer. Besides, based on porous structure, we innovatively introduced a man-made groove which can make adequate dissolution possible in a deeper area, improving the volume ratio of air and decreasing the elasticity modulus of the electric layer further. As for the designed structure, the introduction of pores and the groove increases the porosity and decreases the elasticity modulus to a great extent, and the hemisphere structure makes the applied pressure concentrate on the apex, thus leading to a higher deformation degree and a sharp increase in the effective dielectric constant of the dielectric layer. The sensitivity of the fabricated sensors is relatively high and can be tuned by changing the pore size and the depth of the groove of the elastomer. Compared with the salt template method that has been previously reported, [31] uniformly sized NaCl granules were generated by simply using a group of sifters instead of dissolution and recrystallization. Furthermore, a pin with a diameter of two millimeters was used to form the groove before the curing process without leaving pins to the next step, avoiding the negative influence of shaking the pins before the dielectric layer is cured. The electrodes were fabricated by a simple integration of carbon nanotubes (CNTs) with PDMS. In conclusion, the dielectric layer and electrodes were prepared with soft materials, which makes it possible for the sensor to tightly attach to contact surface with great adaptability. In addition, the fabrication process is simple and low-cost, and the required materials (PDMS, salt, and CNTs) are readily available. The sensor based on the porous, hollow hemisphere dielectric layer showed both high sensitivity and excellent linearity in a wide response range (0.180 kPa^−1^ in the pressure range <50 kPa), which were superior to most sensors of the same kind. The sensor has a low detection limit of 3.5 Pa, good repeatability during the 1000 loading-unloading cycles, and a fast response speed of about 50 ms. Additionally, potential applications such as tiny pressure detection, human-motion monitoring, and object identification of mechanical claw were demonstrated. 

## 2. Methods

### 2.1. Fabrication of the Porous, Hollow Hemisphere Dielectric Layer

PDMS was used for the fabrication of the dielectric layer for its low Young ‘s modulus, high permittivity, and strong corrosion resistance. Figure 1a illustrates a schematic diagram of the fabrication process of the porous, hollow hemisphere dielectric layer (PHHDL). In the first step, a polyvinyl alcohol (PVA) solution was brushed on the 3D hemisphere (radius = 3.5 mm) plate with a cotton to form a sacrificial layer. In the second step, after having been vacuumized a PDMS mixture (PDMS: curing agent = 10:1) was uniformly mixed with a certain size of salts which was selected by sifters with different mesh sizes. In the third step, the compound was poured into the 3D plate, and a hollow column was obtained by a pin with a diameter of 2 mm. Furthermore, a blade was employed to compact the compound. Then, the compound was vacuumized and cured in an oven. Next, the template was immersed with compound in a water tank for a few hours. Finally, the fabricated dielectric layer was peeled off the plate. 

### 2.2. Fabrication of Electrodes

Carbon nanotubes have extremely high strength, and the theoretical calculation value is 100 times that of steel. Besides, carbon nanotubes have a high adsorbability and are very soft. Herein, electrodes were fabricated by integrating CNTs with PDMS. Figure 1b demonstrates the detailed process. Above all, the multi-wall carbon nanotube (MWCNT) suspension was prepared. MWCNT particles, dispersant, and deionized water were fully blended in the ratio of 1:1.6:49, and the mixture was dispersed by ultrasound for 3 h [29]. Secondly, the mixture was transferred into a sprayer and evenly spritzed onto the surface of a clean glass sheet 3 times. In the end, a vacuum-treated PDMS mixture (PDMS: curing agent = 10:1) was shifted onto the aforementioned dried mixture by spin-coating. Due to the strong attraction between CNTs and PDMS, CNTs were absorbed onto the PDMS after being dried, and the conductive electrodes were acquired after being peeled off the sheet. The width of the prepared square electrode is 17 mm and the thickness is 1 mm. Because CNTs in the internal surface of electrodes are toxic, it is necessary to encapsulate the sensor with PU tape when it is used for wearable devices to avoid possible harm to our skin. 

### 2.3. Preparation and Characterization of the Capacitive Pressure Sensor

In order to realize good stationarity, the dielectric layer and electrodes were bound by a silicone sealing adhesive. First, a little adhesive was dipped onto the apex of the hemisphere. Second, the dielectric was put on a prepared electrode. Third, a little adhesive was dipped onto the other side of the dielectric layer and the other electrode was put on it. Figure 1c presents the fabricated porous, hollow hemisphere capacitive pressure sensor, which takes on a typical sandwiched structure (electrode-dielectric layer-electrode) and Figure 1d shows the obtained dielectric layer with a width of 17 mm. The sketch map of the cross-section is shown in Figure 1e, in which blue stands represent the voids produced by dissolved salts and white refers to PDMS. Microscopes were used to clearly observe the inner structure of the PHHDL. Figure 1f–g exhibits the scanning electron microscope (SEM) images of the dielectric layer and Figure 1h–i shows the optical microscope (OM) images of it. As we can see, pores produced by the dissolved salts and the man-made groove are distinctive and relatively regular in the dielectric layer.

## 3. Results and Discussion

### 3.1. Working Principle

In a general way, common capacitors can be regarded as a plane-parallel capacitor where the electrodes are parallel and separated by a dielectric layer, just as Figure 2a shows. As for the flexible capacitive pressure sensor, its working principle can be elucidated as follows: when external pressure stimuli are applied on the sensor, pressure changes result in distance changes between the two electrodes and changes in the relative permittivity of the dielectric, thus leading to capacitance changes [34]. Therefore, information relating to the external pressure can be acquired in real time by recording the output capacitance of sensors. According to the generalized Hooke law [35], the connection between external applied pressure and the distance between two electrodes can be expressed as:(1)$$P=\sigma =\varepsilon E=\frac{\Delta d}{d_{0}}E$$
where *P* and *σ* represent the applied pressure, and *ε*, *d*_0_, Δ*d*, and *E* refer to the relative deformation variable, initial distance, distance variable, and elasticity modulus, respectively. Therefore, when the same pressure is applied to sensors that have identical structures, sensors with a lower elasticity modulus will have a greater relative distance variation. 

The equation of capacitance (*C*) of the capacitive pressure sensor is given by:(2)$$C=\frac{\varepsilon_{0}\varepsilon_{r}A}{d}$$
where *ε*_0_ represents the permittivity of the vacuum and *ε_r_* stands for the relative permittivity of the dielectric layer. Under the application of pressure, the distance between the two electrodes decreases and permittivity increases, thus increasing the capacitance. The definition of sensitivity (*S*) is as follows:(3)$$\;\;S=\frac{\Delta C/C_{0}}{\Delta P}$$
in which *C*_0_ and Δ*C* (=*C – C*_0_) are the initial capacitance and capacitance variation, respectively. According to the above equations, the sensitivity of the capacitive pressure sensor is closely associated to the elasticity modulus and relative dielectric constant. Briefly speaking, the sensitivity is inversely proportionate to the compressive modulus of the elastomer (a larger distance change under a given pressure causes a higher sensitivity), [22] and a more conspicuous increase of dielectric permittivity also improves the sensitivity.

Figure 2b shows our fabricated flexible capacitive pressure sensor based on a porous, hollow hemisphere dielectric layer. Effective relative permittivity of the dielectric (*ɛ_e_*) under the pressure loading is given by:(4)$$\varepsilon_{e}=f_{air}\cdot \varepsilon_{air}+f_{PDMS}\cdot \varepsilon_{PDMS}$$
where *f_air_* is the volume fraction of the air, *f_PDMS_* is the volume fraction of the PDMS (*f_air_* + *f_PDMS_* = 1), *ε_air_* = 1, and *ε_PDMS_* = 2.8. Because the air will be squeezed out and replaced by the PDMS, *ε_e_* will increase overall and then lead to a bigger capacitance. Porous structures have previously been shown to have a lower compressive modulus compared to the bulk of the same material [22]. With regard to the porous structure, deformation occurs via closing of the pores, which requires less pressure. If the pores are open, the sensor will have a relative high sensitivity as the elasticity modulus is low. However, when an overlarge pressure is applied to the sensor with a porous structure, the pores will completely close. Under this circumstance, the mechanical property of the sensor with a porous structure seems to change into that of a solid [11]. However, sensors with a porous structure still have a high sensitivity unless the pressure is considerably high. Besides, the hemispheric structure makes the applied pressure concentrate at the apex of the hemisphere, which makes it easier to deform itself to remove air. In order to decrease the effective compressive further and increase the volume ratio of air to elastomer in the initial state, the man-made groove was introduced [22]. Besides having a lower elasticity modulus, another effect of the as-fabricated structure on the sensitivity is the conspicuous increase of effective dielectric permittivity as applied pressure increases [36]. As the permittivity of air is much lower than that of PDMS, the effective relative permittivity of the dielectric will increase as a given pressure is applied to the fabricated sensor, which accounts for a superior sensitivity than the sensor with bulk material. Figure 2c exhibits the sketch map of the change in d when vertical pressure is applied to the PHHDL-based sensor compared with the sensor without a porous and hollow structure. The distance between the two electrodes of the PHHDL-based sensor reduces dramatically due to its lower compressive modulus, while the non-patterned sensor only has a slight distance decrease. To elaborate on the abovementioned analysis, finite element analysis (FEA) simulations displaying a pressure of 1kPa being applied on the sensors are shown in Figure 2d–k. Figure 2d–g shows the displacement of sensors of non-porous structure, and a void diameter of 300 μm, 450 μm, and 600 μm. Figure 2h–j shows the displacement conditions of sensors with a groove depth of 1 mm, 1.5 mm, and 2 mm with a void diameter of 600 μm. It is evident that displacement conditions of different areas of one sensor are different, which enlightens us that bigger deformations appear at the apex of the hemisphere. By comparing displacement conditions of different sensors, it is easy to find that bigger void diameters and deeper grooves are inclined to promote the deformation of sensors. Therefore, it is meaningful to achieve a higher sensitivity by adjusting relevant dimension parameters. Figure 2k shows a three-dimensional diagram of the PHHDL-based sensor used for our simulations, which clearly shows relevant size information.

### 3.2. Performance of the Sensor

In order to demonstrate the influence of the granule diameter of salt on the performance of sensors, salt particles whose particle diameters were 300–450 μm, 450–600 μm, and 600–700 μm were selected to prepare the porous hemisphere dielectric layer. As we can see in Figure 3a, the porous capacitive sensor fabricated with bigger-sized salt particles shows a higher sensitivity in the pressure region of 0–10 kPa. It seems that larger, opened pores contribute to a lower Young ’s modulus as well as a higher volume ratio of air. Therefore, the distance between the two electrodes will decrease more, and the effective dielectric constant of the dielectric layer will increase to a greater extent as more air is squeezed out, both of which jointly lead to higher sensitivity. Besides the sizes of salt granules, the effect of dissolution time on the relative change in capacitance was also taken into consideration. After being dipped in water for different times (2 h, 4 h, 8 h), porous dielectric layers whose diameters of pores were 600–700 μm were obtained and used to make sensors for further performance comparisons. Stepped pressures were applied from 5 to 50 kPa using a digital pressure gauge, and the corresponding ΔC/C_0_ vs. time plots were shown in Figure 3b. Obviously, as the time of dissolution increased, the fabricated sensor had a more distinct response. What should be noted is that salt particles were already totally dissolved after the dielectric layer was immersed in water for 8 h, since the dielectric layer felt very soft without a solid, hard substance. The experiment powerfully demonstrates the meaning of decreasing residue and increasing pore interconnectivity. To verify the feasibility of grooves and explore the influence of the depth of grooves on the property of the sensor, stresses from 5–50 kPa were applied stepwise every 10 s, and the experimental results are presented in Figure 3c. It is evident that an introduced man-made groove can effectively increase the sensitivity of the sensor and a deeper groove may lead to a more obvious response by decreasing the Young ‘s modulus and increasing the volume ratio of air further. Furthermore, relative changes in capacitance versus time plots of the PHHDL-based capacitive pressure sensor with varied groove depths under a repeated pressure of 350 Pa exerted by a weight of 10 g were shown in Figure 3d. As we can see in the figure, the three sensors had a stable response under the given pressure. Considering all these factors that affect the performance of the prepared sensors, we ultimately selected salt particles with a diameter size of 600–700 μm, prepared a groove of 2 mm deep, and immersed the dielectric layer into water for 8 h to get the optimized PHHDL-based sensor. Then, the prepared sensor was used to carry out the following tests. Pressure–response curves for the optimized fabricated PHHDL-based sensor are depicted in Figure 3e,f. Figure 3e reflects the capacitance response of the sensor in the pressure regime between 0 Pa and 50 kPa, while Figure 3f provides the capacitance response in the low-pressure range, which is 0–35 Pa or so. The abovementioned pressures were applied by a digital pressure gauge and light weights of different mass. Linearity refers to the mathematical relationship or function between the input and output signals of a sensor. A pressure sensor with a high linearity obtains accurate information without additional signal processing or calibration systems [11]. Generally, it is challenging to develop sufficient sensitive pressure sensors with linearity over a wide pressure range due to the trade-off between linearity and sensitivity. For example, the sensitivity of the sensor proposed by Li et al. [34] changes with applied pressure and is found to be 3.13 kPa^−1^ within 0–1 kPa and 1.65 kPa^−1^ within 1–5 kPa, 1.16 kPa^−1^ within 5–10 kPa, 0.68 kPa^−1^ within 10–30 kPa, and 0.43 kPa^−1^ within 30–50 kPa. In comparison, the flexible capacitive pressure sensor prepared by us keeps a better linearity over a wide pressure range (S ≈ 0.18 kPa^−1^ in the pressure range 0–50 kPa). The sensor can even detect a pressure of 3.5 Pa produced by a weight of 100 mg. The capacitance of the sensor changed instantly when a sheet metal of 100 mg was put on and taken away, with a fast response time of about 52 ms and a recovery time of about 56ms, as shown in Figure 3g. As sensors are used under different circumstances, sensors are supposed to possess good resistance to temperature. In order to probe into our sensor’s resistance to temperature, the sensor was tested as the environmental temperature increased from 20 °C to 100 °C. It is worth noting that the capacitance fluctuation was no more than 1%, even though the temperature reached 100 °C. Stability is another important performance parameter of the sensor, and good stability is the precondition for a sensor’s long-time practical application. To ensure the stability of our sensor is necessary, herein, the stability of the sensor was tested by periodic compression using a pressure gauge for 1000 cycles. It can be seen in Figure 3i that the sensor kept steady even if it had gone through repeated compression 1000 times.

Table 1 compares materials and key parameters (including response time and sensitivity) among several capacitive sensors. The prepared pressure sensor using PDMS with pores and groove in this work shows a faster response speed compared with the sensors proposed in Reference [4] and Reference [36], and displays a higher sensitivity than sensors proposed in Reference [35] and Reference [36] and so on, which indicates the performance superiority of the sensor in our work and demonstrates the effectiveness of the designed structure. 

### 3.3. Application of the PHHDL-Based Capacitive Pressure Sensor

In order to illustrate the practicability and availability of our capacitive pressure sensor based on the porous, hollow hemisphere dielectric layer, several tests were carried out, and corresponding test results are shown in Figure 4. Considering a low detection limit and fast response speed with respect to the applied pressure, the proposed sensor is expected to be used to detect tiny pressures. Just as Figure 4a shows, when a lightweight red flower exerting a pressure of only about 30 Pa was repeatedly put on and taken away from the sensor, the sensor had a fast and obvious response. Therefore, our PHHDL-based capacitive pressure sensor could be considered a suitable candidate for detecting tiny pressures. To demonstrate the applicability of our PHHDL-based capacitive pressure sensor for the detection of human motion, the sensor had been encapsulated with a PU tape, which has a low modulus (104~106 Pa) and thickness (5 um), and this was attached to the finger joint and twist of a volunteer with the PU tape. PU tape was selected for the encapsulation and fixation of the sensor due to its transparency, excellent elasticity, and adhesiveness. As shown in Figure 4b, the subject bent the index finger bound with the capacitive sensor from horizontality to 45 degrees and 90 degrees and stayed for 10 s in each state. Then, the subject bent his finger to restore to the original horizontal state. The result shows that the corresponding capacitance change of the attached sensor was stable and it was easy to distinguish the different states of the finger. A similar experiment was conducted on the wrist, just as Figure 4c exhibits. With the degree of bending increasing, more applied pressure on the sensor resulted in a greater distance variation between the two electrodes and a greater permittivity of the dielectric layer, which accounted for the increase in capacitance; as the bend angle decreased, exerted pressure on the pressure sensor reduced, so the capacitance decreased. In view of the fast response speed and high sensitivity of our sensor, pressure information can be readily and accurately acquired by it. These two applications effectively showed the value and potential of our sensor for human motion detection. In the fourth example, the sensor was used to detect small dynamic pressure. Firstly, a light plastic cup was put on the sensor slightly, then a series of liquid droplets were dropped into the cup by a plastic dropper. Figure 4d shows that the capacitance had an obvious stepwise escalation when another drop of water fell into the plastic cup, which indicates that the soft sensor can be used to sense tiny dynamic pressures. Besides the abovementioned applications, the fabricated flexible pressure sensor can also be used together with some tools. For example, a mechanical claw could distinguish objects with different rigidities (e.g., a fresh egg with greater rigidity and a peeled egg with less rigidity) when the pressure sensor was affixed to it (Figure 4d). It seemed that grasping a peeled egg took more time and the corresponding capacitance was smaller. Therefore, it is easy to distinguish objects which have the same size but different rigidities by comparing the capacitance response of the sensor. Furthermore, our sensor was integrated with a mechanical arm to sense the pressure when the mechanical arm clamped a scissor to cut a foam board. Before cutting, the capacitance of the sensor was small because of a low pressure. While the grasped scissor was cutting the board, the foam board was compressed but not cut since it has good condensability, and the capacitance of the sensor increased continuously with the increasing applied pressure. Finally, the foam board was compressed to a limited extent where the board could not be compressed further, and the capacitance of the sensor noted a sharp increase the moment the board was cut off. Based on the relative change in sensor capacitance, information on the cutting conditions could be acquired, which shows that our fabricated sensor has the potential to be used in the machinery field.

## 4. Conclusions

To sum up, the porous capacitance sensor was prepared by means of the “salt template method”, and the difference in the granular size and dissolution time will have an influence on the performance of the sensor. Furthermore, a cylindrical groove was introduced into the architecture to further lower the Young ’s modulus and increase the volume of air, and the results show that the sensitivity did have an obvious increase. The influence of the porous structure and groove was experimentally demonstrated and validated by theoretical analysis and FEA simulation. Finally, a highly sensitive, flexible capacitive pressure sensor was proposed. The sensor has a good linearity over a wide pressure range with a relatively high sensitivity (0.180 kPa^−1^ in the pressure range <50 kPa), and its detection limit is lower than 3.5 Pa. Also, the sensor has a quick response time of about 50 ms, good resistance to temperature, and stable repeatability. Furthermore, the sensor’s practical application value was demonstrated by several tests. The results indicate the potential applications of the sensor in sensing light objects, recognizing gestures, distinguishing objects, and so on. We expect that our flexible senor will be applied for more purposes in the near future. 

## Figures and Tables

**Figure 1 micromachines-14-00662-f001:**
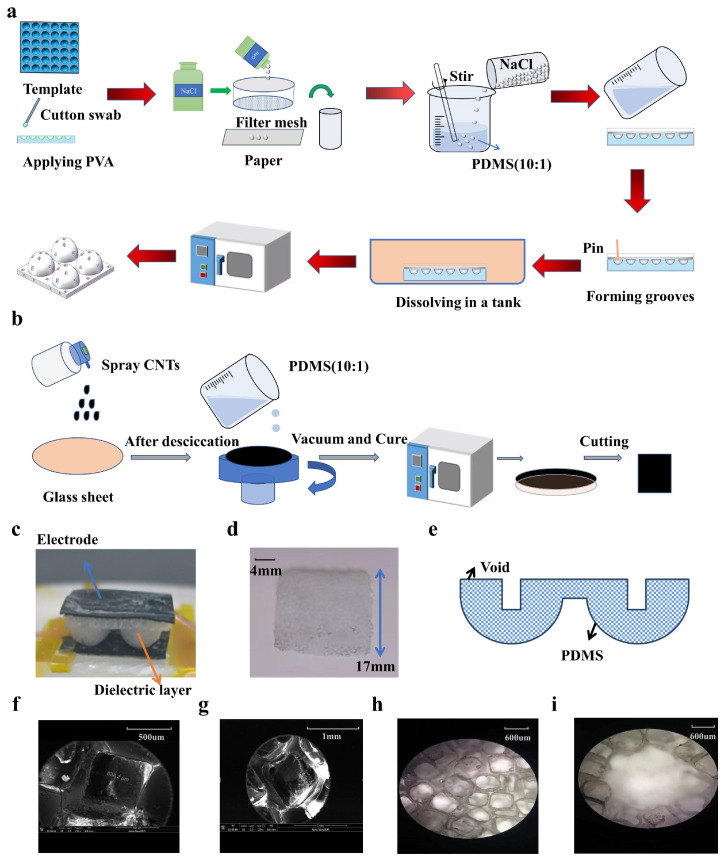
Overview of the capacitive pressure sensor based on a porous, hollow hemisphere dielectric layer (PHHDL). (**a**) Schematic diagram of the PHHDL fabrication process. (**b**) Schematic depiction of the electrode fabrication process. (**c**) The fabricated sensor and dielectric layer. (**d**) The obtained dielectric layer. (**e**) The sketch map of the cross-section of the dielectric layer. (**f**–**i**) Images of the dielectric layer.

**Figure 2 micromachines-14-00662-f002:**
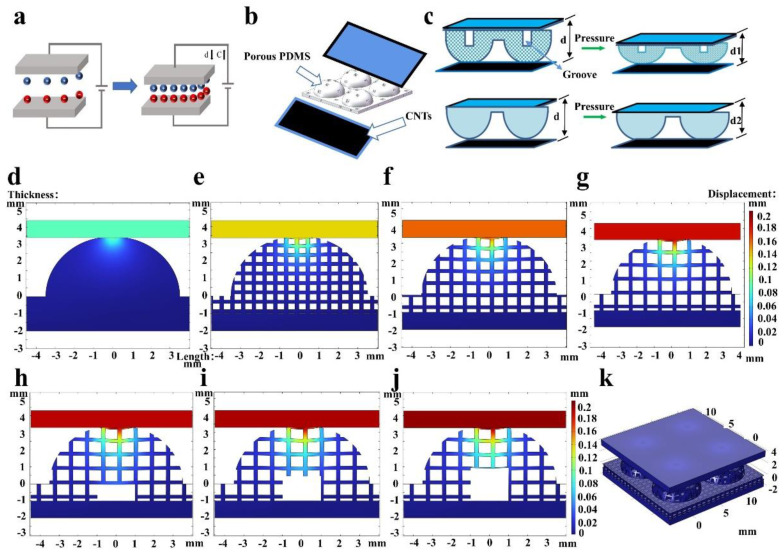
Working principle sketch map and simulation charts. (**a**) Working principle of the plane-parallel capacitive pressure sensor. (**b**) Structure of the PHHDL-based pressure sensor. (**c**) Schematic illustration showing the structure deformation procedure when the dielectric layer is porous, hollow, and non-patterned. (**d**–**k**) Simulation schematics of the pressure sensor when a pressure of 1 kPa is applied to the sensor.

**Figure 3 micromachines-14-00662-f003:**
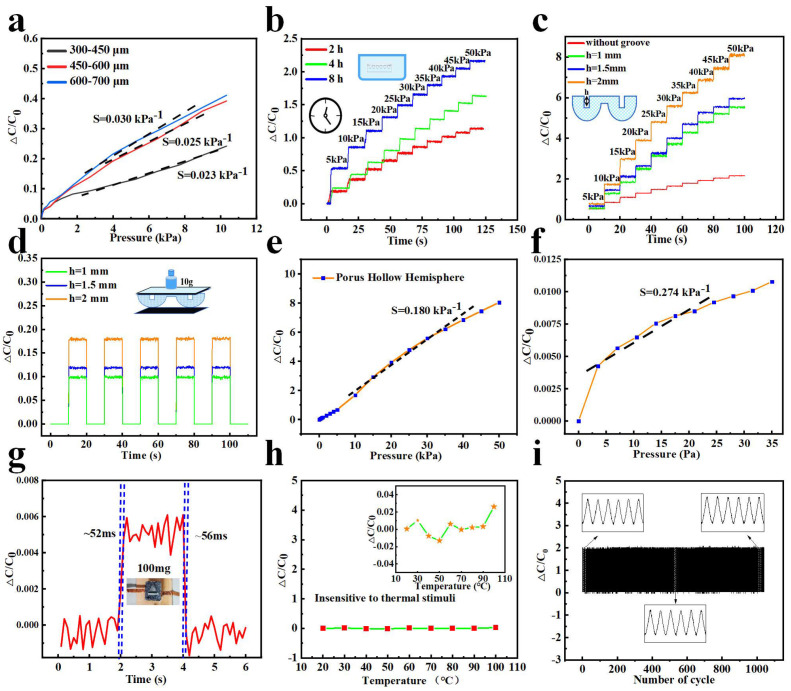
Performance tests of the fabricated sensors. (**a**) The dynamic response of ΔC/C_0_ for the porous hemisphere capacitive sensor based on the salt particles with different diameters (300–450 μm, 450–600 μm, and 600–700 μm). (**b**) The corresponding ΔC/C_0_ for the porous sensors immersed in water for different times (2 h, 4 h, and 8 h). (**c**) The capacitance responses of the pressure sensor based on different groove depths under step-loading pressure circumstances. (**d**) Relative changes in capacitance versus time plots of the PHHDL-based capacitive pressure sensor with varied groove depth under repeated pressure of 350 Pa exerted by a weight of 10 g. (**e**,**f**) Pressure-response curves for the optimized fabricated PHHDL-based sensor. (**g**) Relation and steady-state curve of the pressure sensor with loading and unloading of a weight of 100 mg. (**h**) The stability curve of the flexible pressure sensor with increased temperature. (**i**) Endurance capacitance changes over 1000 cycles at 12 kPa.

**Figure 4 micromachines-14-00662-f004:**
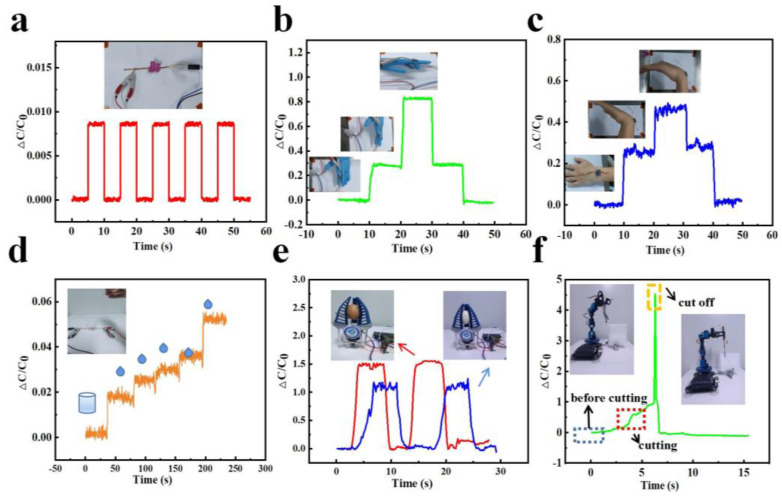
Application tests of the PHHDL-based sensor. (**a**) The tactile sensor repeated real-time responses to a small pressure of ≈30 Pa. (**b**,**c**) Real-time responses of the tactile sensor when the finger or twist joint bent from 0 degrees to 90 degrees and then returned to 0 degrees. (**d**) The dynamic response of ΔC/C_0_ when another drop of water fell into the cup. (**e**) The capacitance changes of the sensor attached to the mechanical claw when the claw grabbed a fresh egg and a peeled egg. (**f**) The capacitance curve of the flexible pressure sensor when the grasped scissor cut a plastic board.

**Table 1 micromachines-14-00662-t001:** Comparison of the materials and key parameters of several sensors.

Dielectric	Electrode	Response Time (ms)	Sensitivity (kPa^−1^)	Ref.
Porous PDMS	Aluminium fabric	100	0.18	[4]
Porous Ecoflex	Conductive fabric	7	0.0121	[36]
Porous PDMS	Conductive cloth tape	155	0.023	[37]
PDMS with pyramidal arrays	Aluminium-coated PET film	<1000	0.15	[24]
Porous PDMS	Aluminum fabric	70	0.102	[38]
Porous PDMS	ITO films	40	0.08	[39]
Porous MWCNTs/BaTiO3/PDMS	Conductive adhesive	≈200	1.24	[40]
Porous PDMS with groove	CNT-coated PDMS	52	0.18	This work

## Data Availability

The data that support the findings of this study are available from the corresponding author upon reasonable request.

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
