# Peer review of "A Sensitive and Flexible Capacitive Pressure Sensor Based on a Porous Hollow Hemisphere Dielectric Layer"

_micromachines, 2023, doi:10.3390/mi14030662_

Round 1
Reviewer 1 Report
In current research a capacitive pressure sensor has been investigated and need to be corrected dramatically.
Minor corrections:
1- Proper us of abbreviation need to be checked such as:
Polyurethane (PU), poly dimethyl siloxane (PDMS)(line 32)
Probably, (PVA) is Polyvinyl Alcohol (line 88)
2- CNT’s are toxic and need to be justified how it will be used in contact with skin.
3- Physical dimension of fabricated electrodes (such as thickness,…) that plays important role on device performance. (Line 106)
4- Figures are not clear enough and need to be re considered.
Major corrections:
1- What is the novelty of paper on fabrication process as salt template method has been reported previously (Reported in https://pubs.rsc.org/en/content/articlelanding/2020/tc/d0tc00443j#!)
2- Conductive electrode fabrication also has been addressed in
https://www.nature.com/articles/s41598-017-18209-w
3- References for part 2.2 need to be added.
4- Equation number 4 is not an acceptable approximation as pressure is more than elasticity modulus (line 133-134) therefore sensitivity analysis based on this approximation is not correct.
5- Line 143 is not right assumption (its ambiguous).
Reviewer 2 Report
In this paper, a sensitive flexible porous capacitive pressure sensor was designed and manufactured by means of “salt template method” and man-made groove. With pores and groove, the sensor is more liable be compressed, which will result in a dramatic decrease of distance between the two electrodes and a conspicuous increase of effective dielectric constant. Besides, the sensor shows a low detection limit as 3.5 Pa and fast response speed (≈50 ms), which makes it possible to detect a tiny applied pressure immediately. The fabricated sensor can be applied in wearable devices to monitor finger and wrist bending, and it can be used in the object identification of mechanical claw and object cutting of mechanical arm and so on. I believe that publication of the manuscript may be considered only after the following issues have been resolved.
1. In order to better highlight the advantages of this work, the author needs to provide a table to compare related work.
2. Some information in Figure 1 needs to be improved by the author. For example, the author in Figure d can add a scale. Figure f-i The scale is not clear enough.
3. The text information in Figure 2 is not clear.
4. The introduction can be improved. The articles related to some applications of flexible pressure sensors should be added such as: Procedia Engineering 47, 2012, 1177-1180; Phys. Chem. Chem. Phys., 2022, 24, 21233; Micromachines 2018, 9(11), 580.
5. Please check the grammar and spelling mistakes of the whole manuscript.
Reviewer 3 Report
In this manuscript, Cui et al. developed a flexible capacitive pressure sensor based on salt template method and man-made groove. The reviewer has the following comments:
The structure of this paper should be improved. A typical structure would be Introduction; Methods; Results and Discussion; Conclusion. The reviewer believes that the fabrication of the sensor and its characterization methods deserve their own section: Methods.
In the last paragraph of Introduction, the authors should highlight the novelty of this paper more clearly, better in bullet points.
Figure 1 (a), the label ‘NaCl’ in the second step should be adjusted.
Figure 1 (f-i), scale bars should be provided and more description should be added in the main text.
Line 133, what are the pressure value and elasticity modulus?
Figure 2 (d-j), what is the meaning of the colorbar? At least a label should be added to it.
Only figure 2&3 have a shaded background. To ensure the consistency of this paper, this background should be removed or added to the rest of the figures as well.
To better demonstrate the novelty of this paper, the authors should compare their design with the existing flexible sensors in terms of sensitivity, response speed, linearity, etc.
English writing requires significant improvements. Proofreading is suggested.
The quality (resolution) of the figures should be improved. As the data plots are relatively small, much information is lost with the current poor resolution.
Round 2
Reviewer 1 Report
Corrected version is improved and comments are addresed.
Reviewer 2 Report
Accept in present form.
Reviewer 3 Report
All the reviewer's comments have been addressed, the reviewer recommends for publication